# The Effects of Periodontal Treatment on Rheumatoid Arthritis and of Anti-Rheumatic Drugs on Periodontitis: A Systematic Review

**DOI:** 10.3390/ijms242417228

**Published:** 2023-12-07

**Authors:** Francesco Inchingolo, Angelo Michele Inchingolo, Pasquale Avantario, Vito Settanni, Maria Celeste Fatone, Fabio Piras, Daniela Di Venere, Alessio Danilo Inchingolo, Andrea Palermo, Gianna Dipalma

**Affiliations:** 1Department of Interdisciplinary Medicine, University of Bari “Aldo Moro”, 70124 Bari, Italy; angeloinchingolo@gmail.com (A.M.I.); avantario@libero.it (P.A.); v.settanni@libero.it (V.S.); dott.fabio.piras@gmail.com (F.P.); daniela.divenere@uniba.it (D.D.V.); ad.inchingolo@libero.it (A.D.I.); giannadipalma@tiscali.it (G.D.); 2PTA Trani-ASL BT, Viale Padre Pio, 76125 Trani, Italy; 3College of Medicine and Dentistry, Birmingham B4 6BN, UK; andrea.palermo2004@libero.it

**Keywords:** cytokines, disease-modifying anti-rheumatic drugs, non-surgical periodontal treatment, oral microbiota, periodontitis, rheumatoid arthritis, scaling and root planning, tumor necrosis factor inhibitors

## Abstract

Rheumatoid arthritis (RA) and periodontitis are chronic inflammatory diseases that widely spread and share the same patterns of pro-inflammatory cytokines. This systematic review aims to evaluate the effects of non-surgical periodontal treatment (NSPT) on RA and, conversely, the impact of disease-modifying anti-rheumatic drugs (DMARDs) on periodontitis. PubMed, Embase, and Web of Science were searched using the MESH terms “periodontitis” and “rheumatoid arthritis” from January 2012 to September 2023. A total of 49 articles was included in the final analysis, 10 of which were randomized controlled trials. A total of 31 records concerns the effect of NSPT on parameters of RA disease activity, including a 28-joint disease activity score, anti-citrullinated protein antibodies, rheumatoid factor, C reactive protein, erythrocyte sedimentation rate, pro-inflammatory cytokines and acute phase proteins in serum, saliva, gingival crevicular fluid, and synovial fluid. A total of 18 articles investigated the effect of DMARDs on periodontal indexes and on specific cytokine levels. A quality assessment and risk-of-bias of the studies were also performed. Despite some conflicting results, there is evidence that RA patients and periodontitis patients benefit from NSPT and DMARDs, respectively. The limitations of the studies examined are the small samples and the short follow-up (usually 6 months). Further research is mandatory to evaluate if screening and treatment of periodontitis should be performed systematically in RA patients, and if the administration of DMARDs is useful in reducing the production of cytokines in the periodontium.

## 1. Introduction

Rheumatoid arthritis (RA) is a chronic, bilateral, and symmetrical inflammatory polyarthritis that mainly affects the synovial membrane of small and large joints and has a deforming and progressive course. Affecting 1% of the world’s population, RA is the most common systemic pathology of autoimmune aetiology [1,2]. 

Periodontitis is a chronic dysbiotic inflammatory disease that affects the supporting structures of teeth, such as the periodontal ligament, root cementum, gingiva, and alveolar bone, representing a major cause of tooth loss [3,4,5]. It is widely distributed in the general population, with a prevalence of 62% [6]. In addition, periodontitis has been identified as a risk factor for the development of systemic diseases such as diabetes mellitus, atherosclerosis, cerebrovascular accidents, cancer, chronic liver disease, myocarditis, and RA [7,8,9,10,11,12]. 

In recent literature, there is increasing evidence to support a bidirectional relationship between RA and periodontal disease, sharing common genetic and environmental factors, such as the expression of the major histocompatibility complex (MHC) class II human leukocyte antigen (HLA)-DRB1 allele, cigarette smoking, and molecular pathways of inflammation [13,14,15]. RA patients present a high prevalence of periodontitis, especially of severe grade (29%) and, conversely, periodontitis patients have a 69% greater risk of developing RA, making periodontitis also a risk factor for the duration of RA [16,17,18,19,20,21,22,23].

It has been supposed that the autoimmunity of RA starts at mucosal sites, such as the lungs, gastrointestinal tract, and oral cavity [24,25]. At this site, the combination of mucosal inflammation and local bacterial dysbiosis could be responsible for triggering the autoimmune response of RA [26,27]. The oral pathogens mainly responsible for periodontitis are identified as “red complex periodontal bacteria”, namely *Tannerella forsythia* (Tf), *Tannerella denticola* (Td), and *Porphyromonas gingivalis* (Pg) [3,28]. DNA of oral bacteria, such as Pg and Prevotella intermedia, was found in synovial fluid [29,30,31]. Models of induced arthritis in mice inoculated with oral pathogens have demonstrated that Pg and *Aggregatibacter actinomycetem comitans* (Aac) can citrullinate host-derived or bacterial proteins [32,33,34]. Aac induces lytic hypercitrullination in host neutrophils through the production of leukotoxin A [35,36]. Pg, through the enzyme peptidyl arginine deiminase, catalyzes the citrullination of type II collagen, resulting in the formation of immune complexes, which lead to the activation of complement and T cells, which, in turn, stimulate B cells to produce anti-citrullinated protein antibodies (ACPAs) [37,38,39,40,41,42,43,44,45,46,47]. APCAs can precede the onset of RA for almost a decade, and a high incidence of periodontitis has been found in APCAs-positive patients without RA [48,49,50]. In addition, Pg alters immune homeostasis in mouse models, enhancing CD19+ B cells, T helper 17 cells, regulatory T cells, and reducing regulatory B cells [51,52,53]. 

RA and periodontitis share the same imbalance of cytokines. Specifically, an up-regulation of pro-inflammatory cytokines, namely interleukine-1 β (IL-1β), interleukine-6 (IL-6), interleukine-8 (IL-8), interleukine-17 (IL-17), tumor necrosis factor-α (TNF-α) and a down-regulation of anti-inflammatory cytokines, namely interleukine-10 (IL-10) and transforming grow factor- β (TGF-β) has been detected both at the level of the synovium and the periodontium also in gingival crevicular fluid [54,55,56,57,58,59]. Specifically, the above-mentioned cytokines promote collagen degradation at both levels [47,60,61,62]. Thus, periodontal inflammation represents a chronic burden of inflammatory cytokines and APCAs, and synovial flogosis constitutes a trap for oral bacteria, contributing each to the induction, maintenance, or exacerbation of the other [62,63,64].

Based on this finding, it can be assumed that the restoration of periodontium could be another key strategy to reduce the severity of RA, just as anti-rheumatic therapy can also positively affect periodontium status [14,65,66,67], in addition to the recent molecular dynamics approach for periodontal treatment [68,69,70]. 

To this purpose, this systematic review aims to provide a comprehensive overview of the effect of non-surgical periodontal treatment (NSPT) on RA disease activity and, conversely, the impact of anti-rheumatic drugs on periodontal indexes to identify potential therapeutic strategies to address both conditions. *Ex iuvantibus*, it could be postulated that RA and periodontitis are sides of the same coin or, better yet, two clinical manifestations of the same autoinflammatory syndrome.

## 2. Materials and Methods

### 2.1. PICO Question

“Do adult patients with RA (P) have clinical and laboratory benefits (O) from periodontal treatment (I) and do adult patients with periodontitis (P) have clinical and laboratory benefits (O) from treatment with anti-rheumatic drugs (I) than the control group (C)?”(Population: adult patients with RA and periodontal disease; Intervention: periodontal treatment or anti-rheumatic drugs; Comparison: control group, patients who have not received periodontal treatment or anti-rheumatic drugs; Outcome: improvement of RA in terms of clinical and/or laboratoristic improvement. Improvement of periodontitis, in terms of reduction of periodontal indexes.)

### 2.2. Protocol and Registration

Our search was performed following the Preferred Reporting Items for Systematic Reviews and Meta-Analysis (PRISMA) guidelines and registered in the International Prospective Register of Systematic Review Registry Guidelines (PROSPERO) (ID: 459633).

### 2.3. Search Processing

The electronic databases PubMed, Scopus and Web of Science were searched to find papers that matched our topic dating from 1 January 2012 to 4 September 2023. The following Medical Subject Headings (MESH) terms were used: “periodontitis” AND “rheumatoid arthritis” (Table 1). 

### 2.4. Inclusion and Exclusion Criteria

Articles were selected if they met the following inclusion criteria: (1) human subjects; (2) English language; (3) randomized controlled trials (RCT), clinical trials, cohort studies, and longitudinal studies; (4) adult patients affected by both periodontitis and RA; (5) patients receiving periodontal treatment or anti-rheumatic medications.

The exclusion criteria were the following: (1) animal models; (2) other languages except English; (3) reviews, case reports, and case series; (4) off-topic article; (5) in vitro studies.

### 2.5. Data Processing

Three reviewers (A.P., V.S., and M.C.F.) independently screened the records according to the inclusion criteria. The reviewers cross-checked each other’s selected articles. Any disagreements were resolved through the arbitration of a senior reviewer (F.I.). The selected articles were downloaded into Zotero (version 6.0.15).

### 2.6. Quality Assessment

The quality of the included papers was assessed by two reviewers, R.F. and E.I., using the ROBINS, which is a tool developed to assess the risk of bias in the results of non-randomized studies that compare health effects of two or more interventions. Seven points were evaluated and each was assigned a degree of bias. A third reviewer (F.I.) was consulted in the event of a disagreement until an agreement was reached.

## 3. Results

### 3.1. Characteristics of Included Articles

A total of 1395 records were identified using the key words “periodontitis” and “rheumatoid arthritis”. When applicable, the automatic filters entered were only human, only in English, only clinical studies, and no reviews. The consulted databases were PubMed (54), Scopus (601), and Web of Science (740). 

During the screening phase, the inclusion and exclusion criteria were applied based on the analysis of the title and the abstract. Only studies that focus on the effect of periodontal treatment on RA and, conversely, on the effect of anti-rheumatic medications on periodontitis were selected. Studies concerning the effects of periodontal treatment on oral health or the effect of anti-rheumatic drugs on RA and studies about the association between RA and periodontitis but focusing on other fields, e.g., epidemiology, aetiopathogenesis, clinical and biological indicators, instrumental examination, and quality of life, were considered off-topic. After screening, 1275 articles were excluded by analyzing the title and abstract, leading to 120 records. Hence, duplicates (57) were manually removed, resulting in 63 records selected. 

After eligibility, 49 studies were included in the analysis, 10 of which were RCT. The process is illustrated in Figure 1. 

### 3.2. Quality Assessment and Risk of Bias of Included Articles

The risk of bias in the included studies is reported in Figure 2. Regarding the confounding bias, most studies have a high risk. The bias arising from measurement is a parameter with low risk of bias. Many studies have a low risk of bias due to bias in participant selection. The bias due to post exposure cannot be calculated due to high heterogeneity. Many studies have low bias due to missing data. The bias arising from measurement of the outcome is low. The bias in the selection of the reported results is high in most studies. The final results show that eight studies have a high risk of bias, two have a very high risk of bias, and four have low risk of bias.

## 4. Discussion

The selected articles were discussed in two sections, the first concerning the effect of NSPT on RA (31 records) and the second concerning the effect of anti-rheumatic drugs on periodontal disease (18 records). Notably, RCTs (10) are included exclusively in the first section.

Generally, the degree of severity of RA and periodontitis are directly proportional in terms of major extension, severity, and occurrence [86]. It has been demonstrated that all periodontal indexes are worse in RA patients, with an adjusted odds ratio of 2.66 [87]. There is also a linear correlation between clinical and laboratoristic parameters of RA and periodontitis, such as the high Disease Activity Score Calculator for Rheumatoid Arthritis (DAS28), RF, ACPAs, CRP, ESR, and disease duration [88,89,90,91].

Elevated serum levels of IgG anti-Aac and Pg have been detected in patients with elevated serum levels of APCAs. They are associated with a poor therapeutic response quantified with DAS28 [32,38,92,93]. Conversely, low serum levels of IgG anti-Pg peptidyl arginine deiminase (PPAD) allow greater clinical response to anti-TNF agents in RA patients, significantly reducing DAS28 and APCA levels [35,94,95]. Therefore, serum levels of IgG anti-Pg can be used as a predictor factor for the clinical response of RA patients treated with biological therapies [96].

### 4.1. The Effect of Non-Surgical Periodontal Treatment (NSPT) on Rheumatoid Arthritis (RA)

Patients enrolled in various studies underwent NSPT, consisting of scaling and root planning (SRP), good oral hygiene instruction, and, in some cases, the administration of systemic antibiotics [97,98]. No patient underwent surgical periodontal treatment. The parameters considered to evaluate the efficacy of NSPT are the following: clinical and instrumental scales, general state of health perceived by patients, levels of systemic inflammation indices, autoantibodies, pro-inflammatory cytokines, and other inflammatory mediators. In no study was the patients’ underlying therapy changed; therefore, it cannot be excluded that the improvement in parameters is related to the therapy for RA. The follow-up during NSPT varies from 45 days to 6 months. Table 2 summarizes the studies examined.

Several authors agreed that NSPT treatment significantly reduces DAS28. The DAS28 algorithm measures disease activity and remission of RA by combining for each patient the number of tenderness and/or swelling of 28 joints, an index of inflammation, and the patients’ global assessment of their health on a 10 cm visual analogue scale (VAS) [127]. As an index of inflammation, the erythrocyte sedimentation rate (ESR) or, more recently, the C-reactive protein (CRP) can be used (DAS28-ESR and DAS28-CRP, respectively). Usually, DAS28-ERS cut-offs are higher than the corresponding DAS28-CRP values [128]. In all the studies, the reduction of DAS28, evaluated at a maximum follow-up of 6 months, correlated with the improvement of periodontal parameters [86,98,100,103,104,105,106,113,116,122]. In the RCT of Ortiz P. et al., a reduction of DAS28 was already detected after 6 weeks following NSPT treatment [98]. Moura M.F. et al. added that the significant improvement of DAS28 in RA patients who underwent NSPT is parallel to the reduction of subgingival bacterial levels of Aac, Pg, Tf, Td, the main agents of periodontitis [70]. These data are confirmed by Cosgarea R. et al., according to which DAS28 correlated positively with those of Pg and negatively with the plaque index [102]. 

Other authors evaluated the RA disease activity with DAS28-CRP. According to the RCT of Nguyen VB et al., the combination of oral hygiene instructions and SRP significantly reduced DAS28-CRP after 6 months of follow-up [103]. However, six months after the treatment, the differences between the treatment group and the control group have already been canceled, revealing that periodontal treatment must be continuous to reduce the activity of RA [103]. Okada detected the reduction of DAS28-CRP after 8 weeks [113]. De Pablo P. et al. demonstrated that after 6 months of NSPT, not only the DAS28-CRP score was reduced but also radiological severity scales, namely the Ultrasound grey scale and power Doppler scores, obtained from all metacarpo-phalangeal joints and dorsal wrists on both sides [105]. Nonetheless, the authors highlighted that it is mandatory to keep the patients motivated to consistently undergo periodontal treatment [105]. Biakowa K. et al. observed that NSPT significantly reduced the disease activity of RA, measured by DAS28-CRP, but did not affect the disease index of ankylosing spondylitis [106]. 

Contrasting data derives from other studies. The most consistent and longer study about the effect of NSPT on RA is the RCT of Mariette S., which revealed that NSPT, consisting of scaling and polishing twice a year for 24 months, did not affect RA activity, measured by DAS28-ESR, although it reduced the prevalence of “red complex periodontal bacteria” in subgingival pockets [109]. According to the RCT “ESPERA” (Experimental Study of Periodontitis and Rheumatoid Arthritis), after a NSPT consisting of SRP and systemic antibiotics, there was no evidence of improvement in DAS28-ESR at 3 months, as well as according to the American College of Rheumatology 20%, 50%, and 70% improvement criteria, the general Health Assessment Questionnaire, and General Oral Health Assessment Index (GOHAI) scores. It is noticeable that a positive trend in the “psychological impacts” domain of GOHAI was found [107]. Similar results derive from the clinical study of Posada-Lòpez A: NSPT has a positive effect on only quality of life and self-reported health indicators (Oral Health Impact Profile-14 and Short Form Health Survey 36), ameliorating psychological distress, emotional role, and mental health. No significant changes are reported in DAS28 [108]. 

In the study of Kaushal S. et al., instead of DAS28, the Simple Disease Activity Index (SDAI) was assessed as an index of disease activity. The SDAI is a simple-to-use scoring system that focuses on the main symptoms of RA, combining the results of five clinical and laboratory items [110]. 

Systemic indices of inflammation, namely CRP and ESR, correlate with the degree of activity of RA [129]. Several above-mentioned authors agreed that CRP and ESR are significantly reduced at 3–6 months after NSPT [98,100,101,102,103,104,105,115,116,122,130]. 

Conversely, according to other authors, NSPT did not significantly affect the levels of ESR and CRP at 3 months [107,110,111]. Paradoxically, the study by Botero et al. found an increase in CRP at 3 months after NSPT [112]. 

The diagnosis of RA is corroborated by the presence of autoantibodies in the serum, such as rheumatoid factors (RF) and ACPAs [131]. The sensitivity and specificity of ACPAs and RF for the diagnosis of RA are 66.0%, 90.4%, 71.6%, and 80.3%, respectively [132]. ACPAs have the highest positive predictive value for the development of RA and are correlated with a poor prognosis [133]. 

Zhao X. et al. and Nguyen V.B. et al. demonstrated that SRP can decrease the level of serum ACPAs at 3 and 6 months, respectively [103,104]. According to the RCT of Okada M. et al., supragingival scaling without root planing is sufficient to reduce the level of ACPAs and serum levels of IgG to Pg hemin binding protein 35 at 8 weeks [113]. Analogue results derived from the comparative study of Lappin D.F. et al., who also observed that smoking habits have no influence on ACPAs in periodontitis patients, and from the observational study of Yang N.Y., according to which the reduction of ACPAs correlates significantly with the number of extracted teeth [99,114]. In a few studies, NSPR was ineffective or, paradoxically, increased the titer of ACPAs [101,110,115]. 

In most studies, the reduction in RF does not reach the level of statistical significance [29,42,43,51,57]. Some authors reported a reduction in FR after NSPR [101,112,124]. These data corroborate the specific role of Pg in inducing autoantigens citrullination through the conversion of arginine to citrulline by the enzyme peptidyl arginine deiminase (PAD) at the synovia level. 

NSPT affects the levels of proinflammatory cytokines and various inflammatory mediators in serum, gingival crevicular fluid (GCF), saliva, and synovial fluid in RA-patients (Table 3). 

In serum, reduction of the following cytokines was observed: TNF-α, IL-6, receptor activator of nuclear factor-KB ligand (RANKL), carbamylated protein (CarP), neutrophil extracellular traps (NETs), the apoptosis inhibitor survivin, and the neuroendocrine hormone prolactin [70,98,99,100,115,117,118,119]. RANKL is expressed by stromal cells and osteoblasts to regulate osteoclastogenesis; its dysregulation has been correlated with osteoporosis, bone cancer, and RA [134]. The protein CarP has recently been identified as a novel autoantigen in RA patients, in whom anti-CarP antibodies are present in 36–40% and are predictive of joint erosions [135]. NETs are produced as a result of innate immunity through oxygen-reactive species and are involved in defense mechanisms against big microorganisms [136]. The dysregulation of NETs has been correlated with several immune-mediated diseases, including RA, because they promote inflammation and thrombosis [137]. CarP and NETs significantly correlated positively with the mean values of probing depth, the clinical attachment level, and the severity of periodontitis. 

Prolactin is a neuroendocrine hormone that mainly regulates lactation but also acts as an inflammatory cytokine, being involved in the pathogenesis of several inflammatory diseases, such as SLE, multiple sclerosis, cancer, and periodontitis [138]. Survivin is a protein that regulates apoptosis and micro-RNA, but also the maturation of B cells and the imbalance of regulatory B and T cells, being involved in the pathogenesis of several immune-mediated conditions, such as RA [139]. 

In GCF, NSPT reduces the levels of TNF-α, IL-1β, IL-6, matrix metalloproteinase-8 (MMP-8), prostaglandin E2 (PGE2), tissue plasminogen activator (t-PA), plasminogen activator inhibitor-2 (PAI-2), and prolactin [117,119,120,121,122]. Affecting the homeostasis of extracellular matrix and cartilage degradation, the enzyme MMP-8 increases the severity of RA [140]. PGE2 is a lipid inflammatory mediator generated also by chondrocytes and synovial fibroblasts, which exerts both pro-inflammatory and anti-inflammatory effects in RA [141]. Recent evidence shows that dysfunction of the t-PA/PAI system contributes not only to thrombosis but also to the pathogenesis of connective tissue diseases and arthritis [142]. 

In the synovial fluid, RA patients undergoing NSPT present a reduction of prolactin [119]. At the salivary level, a reduction of RANKL was detected [70].

It has been proven that RA patients have an increased cardiovascular risk due to the evolution of the disease and the side effects of GCs and non-steroidal anti-inflammatory drugs (NSAIDs) [143]. The relative risk of cardiovascular events in RA is 1.48, while the relative risk of mortality due to a major cardiovascular event is 1.50 [144]. Most deaths in RA are linked to a major cardiovascular event related to atherosclerosis and heart failure. The common pathogenetic mechanisms underlying the association between cardiovascular diseases, RA, and periodontitis are the following: alteration of cytokine levels by bacterial endotoxins, oxidative stress, cross-reaction between RA-specific autoantibodies and antibodies against Pg on cardiac cells, endothelial dysfunction, and alteration of the lipid balance [145]. Salah S. et al. demonstrated that NSPT for 6 months reduced the carotid intimal medial thickness of RA patients and tends to reduce the Framingham risk score, exerting a cardiovascular protective effect [123]. 

Finally, in addition to the mechanical removal of the plaque through SRP, natural compound and photodynamic approaches have been tested in RA patients as adjuvant treatments [146,147,148]. Curcumin, the biologically active part of the *Curcuma longa* root, belongs to the phenolic compound group and has demonstrated antioxidant and anti-bacterial properties. In fact, in vitro studies have shown that curcumin reduces the inflammatory cascade through inhibition of the caspase-1 pathway, toll-like receptor-4, MyD88, nuclear factor kappa light chain enhancer of activated B cells, and reactive oxygen species production. Furthermore, it promotes the osteogenic differentiation of mesenchymal stem cells of the periodontal ligament [149,150]. A mouthwash containing essential oils (clove oil, eucalyptol, thymol, and tea tree) and curcumin in addition to SRP has proven to be effective in reducing ESR, CRP, ACPAs, and RF at 6 weeks, more than SRP alone and SRP associated with a chlorhexidine mouthwash as an adjuvant [124]. The local application of curcumin could be an effective adjuvant treatment for RA patients with periodontitis. 

The use of a methylene blue-based photosensitizer (MB—Sigma-Aldrich, Riyadh, Saudi Arabia) and, subsequently, a HELBO^®^ TheraLite–diode laser (Bredent Medical, Senden, Germany) within the periodontal pockets in addition to SRP significantly decreased the levels of IL-6 and TNF-α in patients with RA and periodontitis at 6-weeks more than SCR alone [125]. No changes were observed in the RF level in GCF. In addition, the same devices can also reduce markers of oxidative stress in GCF, such as 8-hydroxy-2′-deoxyguanosine and 4 hydroxynonenal, which derive from DNA damage and lipid peroxidation, respectively [126]. A photodynamic laser treatment can enhance the mechanical effect of SRP in reducing a source of local inflammation in RA patients. The photosensitization process triggers a phototoxic reaction which allows selective destruction of pathological tissues.

### 4.2. The Effect of Anti-Rheumatic Drugs on Periodontal Disease

The most recent guidelines for the treatment of RA (EULAR task force of 2013) suggest a “treat-to-target” approach using disease-modifying anti-rheumatic drugs (DMARDs), molecules capable of slowing down the progression of joint damage by modulating the levels of cytokines underlying the chronic inflammation [131]. DMARDs are divided into two groups: synthetic disease-modifying anti-rheumatic drugs (sDMARDs) and biologic disease-modifying anti-rheumatic drugs (bDMARDs). The most used conventional sDMARDs recommended during the first lines of therapy in association with glucocorticoids (GCs) are the following: methotrexate (MTX), hydroxychloroquine (HCQ), leflunomide (LFN), and sulfasalazine (SSZ). Tofacitinib and baricitinib, oral Janus-activated kinase (JAK)-dependent cytokine signaling inhibitors, are classified as “targeted DMARDs”. GCs, such as prednisone or equivalent, can be considered as sDMARDs and are used at low doses in the in the early stages to obtain a rapid clinical response and slow down the progression of joint damage, waiting for the DMARDs to exert their effects. sDMARDs can be used in monotherapy or in combination therapy with other sDMARDs or with a bDMARDs. bDMARDs include TNF-α inhibitors (adalimumab, certolizumab pegol, etanercept, golimumab, and infliximab), the T-cell costimulatory inhibitor (abatacept), the anti-B-cell agent, rituximab, the IL-6 receptor blocking monoclonal antibody (tocilizumab), and the IL-1 inhibitor (anakinra) [151]. RA patients affected by periodontitis are less responsive to biological treatments because the inflamed periodontium is a further source of pro-inflammatory cytokines [152,153]. Until recently, it seemed obvious that the use of immunosuppressive drugs increased the risk of periodontitis, as with any type of local or systemic infection [154]. The most recent scientific evidence proves exactly the opposite.

The parameters considered in the selected studies to evaluate the effect of anti-rheumatic drugs on periodontitis are both clinical and immunological. The clinical parameters include periodontal indexes, for example the number of missing teeth, plaque index (PI), gingival index (GI), probing depth (PD), clinical attachment level (CAL), and bleeding on probing (BOP). The immunological parameters consist of the evaluation of specific cytokine levels in saliva or GCF. The studies concerning the effect of sDMARDs and bDMARDs on periodontitis are schematized in Table 4 and Table 5, respectively. 

Kaczyński T. et al. showed that RA patients with periodontitis taking sDMARDs (MTX and LFN) with or without GCs presented reductions in BPO and PD. They also detected lower salivary levels of IL-6, IL-8, IL-17A, and TNF-α in both periodontitis patients without RA and healthy controls. No differences were found between the effects of different medications [71].

Jung G.U. et al. showed that PD and CAL improved in RA patients with periodontitis treated with MTX, HCQ, and SSZ for 4 weeks compared to the control group (healthy subjects with periodontitis). The authors chose to carry out such a short-term follow-up to avoid the effect of patients learning good oral hygiene rules by evaluating only the effect of sDMARDs. No significant difference in periodontal parameters was found for combination therapies of multiple sDMARDs and for the addition of NSAIDs and/or GCs [72]. Nik-Azis N.M. et al. compared two groups of patients with periodontitis, the first with RA and the second with osteoarthritis. They observed that MTX in monotherapy reduction in PD at 3 months, compared with MTX in combination with SSZ and HCQ [73].

The JAK-inhibitor baricitinib has been approved for the treatment of moderate-to-severe RA [155]. In RA patients, baricitinib improved periodontal conditions after 24 weeks in terms of improved GI, sites with BOP, PD, and sites with PD ≥ 4 mm [85]. 

Among bDMARDs, the anti-TNF alpha monoclonal antibodies are the most used for the treatment of RA [156]. Their effect on periodontium is due to the inhibition of pro-inflammatory cytokines and bone resorption, as was demonstrated by Puncevicene E. et al., who reported a reduction of bone loss after administration of bDMARDs with or without sDMARDs [16]. 

Patients treated with adalimumab showed a significant decrease in GI, sites with BOP, and PD [95]. Infliximab improved BPO, PD, and CAL, as well as TNF-α in GCF in patients affected by RA and periodontitis, compared with RA patients treated without infliximab and healthy controls [55,78]. A minor effect was found after treatment with etanercept, limited to BOP, GI, and TNF-α levels in GCF [78]. Golimumab, as well as etanercept and adalimumab, not only exerted a significant effect on periodontal indexes, but also improved periodontal condition in terms of extension and severity of periodontitis, after a follow-up after more than 12 months [77]. 

Finally, anti-TNFα antibodies can exert their effect also in biochemical parameters of periodontium, namely salivary IL-8 and monocyte chemoattractant protein-1 levels and GCF-TNF-α [76,79]. 

The main non-anti-TNF bDMARDs used for RA are rituximab, a chimeric monoclonal antibody targeted against CD20 on B cells, and tocilizumab, an anti-IL6 receptor inhibitor [157,158]. Both rituximab and tocilizumab ameliorated different periodontal indexes. Rituximab was the only bDMARD for which a longer follow-up was carried out—in this case, 48 months—with evidence of clinical improvement in the periodontal status, as well as the periodontal indices [83]. 

According to Kobayashi T. et al., tocilizumab improves periodontal parameters at 6 months equally like anti-TNF drugs, with the difference that it also has positive effect on CAL. This improvement is probably correlated to the reduction of serum inflammatory mediators, namely TNF-*α*, total immunoglobulin G, and serum amyloid A [75,84]. Ancuta C. et al. detected that tocilizumab exerted its effect at 3 months on several periodontal indexes, and that treatment for almost 6 months is necessary to treat to reduce PPD [85]. Figure 3 represents the bidirectional effect of bDMARDs on the therapeutic targets shared both by periodontitis and RA.

In contrast to this evidence, other authors refuted the positive effect of anti-rheumatic drugs on periodontitis or, paradoxically, described a worse periodontal status after anti-rheumatic therapy. The reason lies in the fact that RA patients undergoing immunosuppressive treatment have a greater risk of developing infections, including periodontitis. In a prospective follow-up study carried out in Finland, Äyräväinen L. et al. refuted this statement; during a 6-month follow-up, there were no significant changes in the periodontal parameters of the enrolled patients compared to healthy controls after 6 months of anti-rheumatic treatment, using sDMARDs in monotherapy or in combination with another sDMARDs or a bDMARDs. Likewise, the presence of Pg is also unchanged between the two groups [82]. The same author detected that the administration of sDMARDs and bDMARDS did not affect the levels of salivary MMP-8, an enzyme produced by neutrophils that degrades type I collagen in the periodontium and type II collagen in articular cartilage, representing a new biomarker in periodontitis [81]. According to de Smit M.J. et al., periodontal inflamed surface area did not improve after treatment with MTX for 2 months or anti-TNF antibodies for 3–6 months vs. baseline, as did the level of anti-Pg antibodies [80]. 

Paradoxically, Ziebolz D. et al. found that the DMARDs that most negatively affect the state of periodontium in RA patients are the combination of MTX and anti-TNF-α, compared to LFN in monotherapy and to the combination of MTX and rituximab [81].

The limitations of this study are the following: the small samples, the variability of the follow-up (usually 6 months), the heterogeneity of the periodontal parameters examined, and the small number of RCT. Therefore, it is difficult to establish the optimal frequency with which NSPT should be performed in RA patients and identify the most effective anti-rheumatic drugs for periodontal inflammation. 

## 5. Conclusions

Periodontitis represents a modifiable risk factor for RA, triggering and maintaining immune-mediated inflammation. A simple, repeatable, and low-cost procedure such as NSPT can contribute to extinguishing the inflammatory cascade, sparing the use of systemic pharmacological treatments for RA and avoiding life-threatening complications, primarily cardiovascular events. Chemical mouthwash with natural compounds and photodynamic treatment are being tested as adjuvants to SRP. The analysis of the studies reveals that NSPT seems to have a positive effect more on the reduction of clinical disease activity indices than on the reduction of ESR and CRP, probably because the latter are influenced by various factors, primarily infections, which are higher in rheumatic patients, chronically subjected to GCs and immunosuppressive treatment. According to our results, periodontal examination should be routinely included in RA therapy guidelines to detect and treat periodontitis at an early stage. 

On the other hand, this systematic review demonstrates ex iuvantibus that periodontal disease and RA influence each other via cytokine production and activation of humoral and cell-mediated immunity. Although there are conflicting data in the literature, several authors have demonstrated that treatment with DMARDs also exerts positive effects on the clinical and immunological parameters of the periodontium, corroborating the hypothesis of a bidirectional pathogenetic link between RA and periodontitis. Further studies will be necessary to establish whether, in terms of cost–benefit, it is worth using bDMARDs also in the treatment of periodontitis, considering the high costs and the infectious and neoplastic risks.

## Figures and Tables

**Figure 1 ijms-24-17228-f001:**
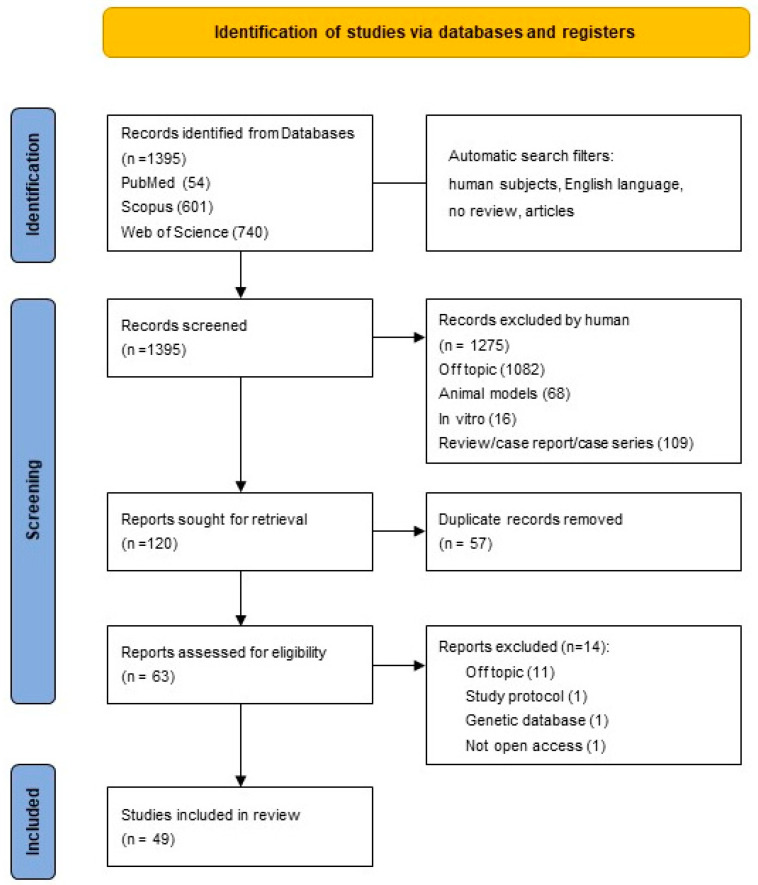
Preferred Reporting Items for Systematic Reviews and Meta-Analysis (PRISMA) flow-chart.

**Figure 2 ijms-24-17228-f002:**
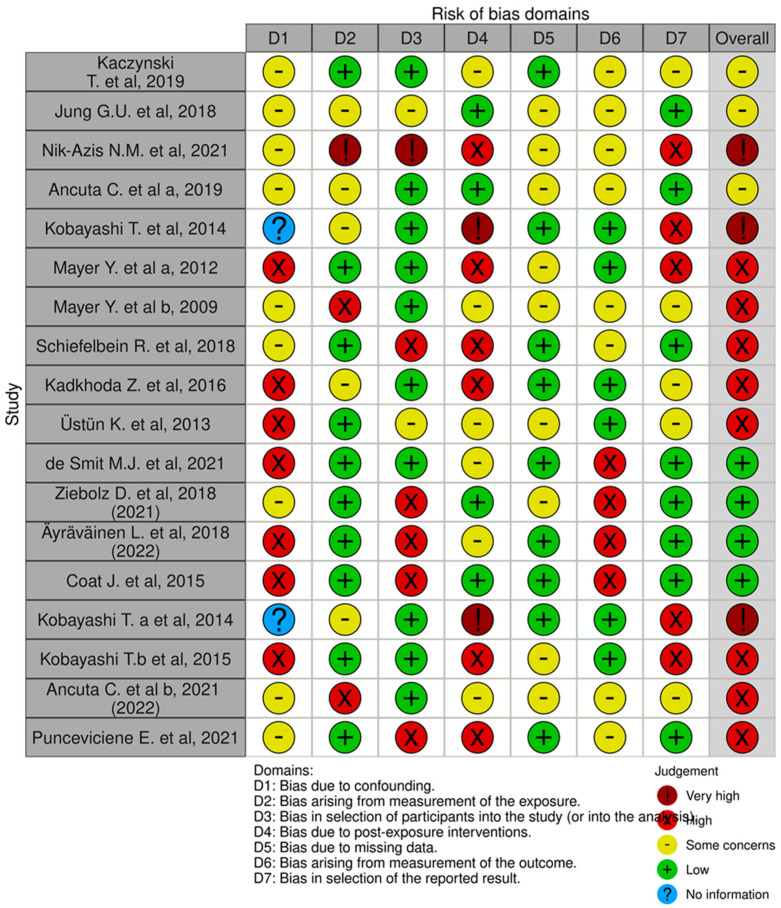
Bias assessment evaluated by Robins, references [16,71,72,73,74,75,76,77,78,79,80,81,82,83,84,85].

**Figure 3 ijms-24-17228-f003:**
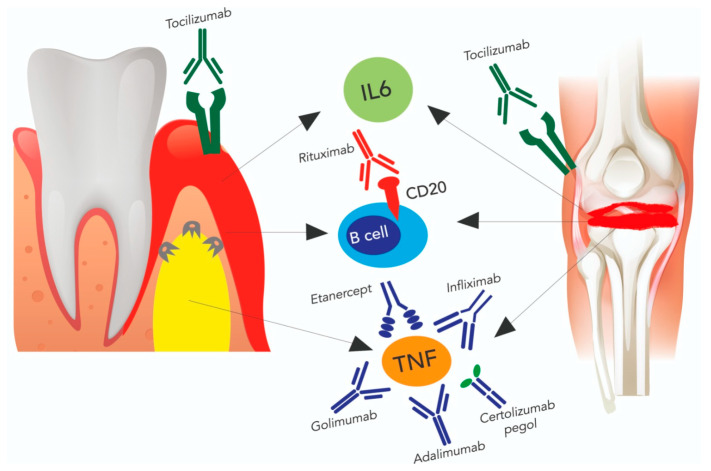
The bidirectional effect of biological disease-modifying anti-rheumatic drugs (bDMARDs) on common therapeutic targets of RA and periodontitis. [IL-6: interleukine-6, TNF: tumor necrosis factor].

**Table 1 ijms-24-17228-t001:** Article screening strategy.

Articles screening strategy	MESH terms: A: “rheumatoid arthritis”; B: “periodontitis”
	Boolean Indicators: “A” AND “B”
	Timespan: from 1 January 2013 to 4 September 2023
	Electronic Database: Pubmed, Scopus, and Web of Science

**Table 2 ijms-24-17228-t002:** Studies concerning the effect of non-surgical periodontal treatment (NSPT) on rheumatoid arthritis (RA).

Authors	Study Design	Study Sample (N.)	NSPT	Follow-Up	RA Parameters	Outcomes
Ortis P. et al., 2019 [98]	RCT	RA + P (20) RA + P (20)	SRP	6 we.	DAS28, ESR, serum TNF-α	Reduction of DAS28, ESR, serum TNF-α
Yang N.Y. et al., 2018 [99]	Observational prospective	RA + P (31)	FMS and subgingival RP	8 we.	ACPAs, RF, CRP, serum IL-1β, IL-6, TNF-α	Reduction of ACPAs and TNF-α. Positive correlation between n. of extracted teeth and reduction of ACPAs and IL-1β levels vs. baseline
Erciyas K. et al., 2013 [100]	Observational prospective cohort	RA + P (60)	SS and RP	3 mo.	DAS28 CRP, ESR, serum TNF-α	Reduction of CRP, ESR, DAS28, TNF-α vs. baseline
Thilagar S. et al., 2022 [101]	RCT	RA + P (13) RA + P (15)	SS and RP	8–12 we.	DAS28 ACPAs, RF, CRP	Reduction of DAS28 and CRP in intervention group Increase of APCAs and RF
Moura M.F. et al., 2021 [86]	RCT	RA (23) RA + P (24) P (4) Healthy (30)	SRP	45 days	DAS28 Bacterial sampling	Reduction of DAS28, Aac, Pg, Tf, Td in RA + P group
Cosgarea R. et al., 2018 [102]	Prospective CC	RA+ P (18) P (18)	SRP	3–6 mo.	DAS28, CRP, ESR, RF Bacterial sampling	Reduction of CRP in RA + P group Reduction of Pg, Tf, and Td in P group Positive correlation between Pg levels and DAS28
Nguyen V.B. et al., 2021 [103]	RCT	RP + P (41) RP + P (41)	SS and RP	3–6 mo.	DAS28-CRP, ACPAs, RF, ESR, CRP	Reduction of DAS28-CRP, ESR, ACPAs in intervention group
Zhao X., 2018 [104]	Prospective	RA (18) RA + P (18) P (18)	FMS and RP	1 mo.	DAS28-CRP, ACPAs, RF, ESR, CRP	Reduction of DAS28, CRP, ESR, ACPAs in RA + P group
De Pablo P. et al., 2022 [105]	RCT	RP + P (30) RP + P (30)	SRP	3–6 mo.	DAS28-CRP, ESR, CRP, musculo-skeletal ultrasound grey scale and power Doppler scores	Reduction of DAS28-CRP, musculo-skeletal ultrasound grey scale and power Doppler scores in intervention group
Białowąs K. et al., [106]	CC	RA + P (44) SpA + P (30) Healthy (39)	SRP	6 we.	DAS28-CRP, BASDAI, ESR, CRP, TNF-α, MMP-3, MMP-9 Bacterial sampling	N.S.C. in detection of Pg Reduction of DAS28-CRP in RA + P group
Monsarrat P. et al., 2019 [107]	RCT	RP + P (11) RP + P (11)	FM and RP, systemic antibiotics	3 mo.	DAS28-ESR ACR20, ACR50, ACR70 criteria, HAQ, GOHAI scores	N.S.C. Positive trend in the “psychological impacts” domain of GOHAI in intervention group
Posada-López A. et al., 2022 [108]	Clinical trial	RA + P (29)	FM and RP	3 mo.	DAS28 SF-36, OHIP-14	N.S.C. in DAS28 Reduction of psychological distress, emotional role, and mental health in intervention group
Mariette X. et al., 2020 [109]	RCT	RA + P (238) RA + P (234)	S and polishing twice a year	24 mo.	DAS28-ESR Bacterial sampling	N.S.C. in DAS28-ESR Reduction of Pg, Tf, and Td in intervention group
Kaushal S. et al., 2019 [110]	Controlled trial	RA + P (20) RA + P (20)	FM and RP	8 we.	SDAI ACPAs, CRP, ESR	Reduction of SDAI in intervention group N.S.C. in ACPAs, CRP, and RF
Roman-Torres C.V. et al., [111]	Longitudinal	RA + P (12) P (12)	S	3 mo.	CRP, ESR	N.S.C. CRP and ESR
Botero J.E. et al., 2021 [112]	Prospective	RA + P (29) P (21)	FM and RP	3 mo.	ACPAs, CRP, RF	Elevation of CRP Reduction of ACPAs and RA in RF group
Okada M. et al., 2013 [113]	RCT	RA + P (26) RA + P (29)	SS	8 we.	DAS28-CRP, ACPAs, IgG anti-Pg	Reduction of ACPAs, DAS28-CRP, and IgG anti-Pg in intervention group
Lappin D. et al., 2013 [114]	Comparative	RA + P (26) RA + P (29)	SS and RP	6 mo.	ACPAs, IgG anti-Pg	Reduction of ACPAs and IgG anti-Pg in intervention group
Ding N. et al., 2022 [115]	Observational cross-sectional	RA + P (32) P (29) RA (7) Healthy (20)	SS, subgingival scraping, RP	6 we.	ACPAs, IL-6, CRP, ESR, RF	Reduction of ESR, CRP, and IL-6 in the RA + P group N.S.C. in APCs and RF
Khare N. et al., 2016 [116]	CC	RA + P (30) P (30)	SRP	3 mo.	DAS28, VAS, ESR, CRP	Reduction of DAS28, VAS, ESR, CRP in RA + P group
Balci Yuce H. et al., 2017 [117]	Controlled trial	RA + P (17) P (18) Healthy (18)	SRP	6 we.	Vitamin D, TNF-α, RANKL, in GCF and serum	Reduction of serum RANKL and GCF TNF-α in RA + P group
Kaneko C. et al., 2018 [118]	Retrospective CC	RA + P (40) P (30) Healthy (43) RA + P (22) treated	FMS	2 mo.	Serum CarP, NETs	Reduction of CarP and NETs in RA + CP group
El-Wakeel N.M. et al., 2023 [119]	RCT	RA + P (20) P (20) RA Healthy	SRP	3 mo.	PRL in serum and GCF, ESR, DAS28	GCF and serum levels of PRL in both P group, reduction of synovial fluid PRL in RA + P group N.S.C. in DAS28-ESR
Kurgan Ş. Et al, 2016 [120]	Observational	RA + P (27) P (26) Healthy (13)	FM and RP	3 mo.	GCF MMP-8, PGE2, IL-6, DAS28	Reduction in GCF-MMP-8, PGE2 and IL-6 in RA + P group
Kurgan S. et al., 2017 [121]	Prospective	RA + P (15) P (15) RA (15)	FM and RP	3 mo.	DAS28, CRP, ESR, t-PA and PAI-2 in GCF	Reduction of GCF-t-PA and PAI-e in RA + P group, reduction of CRP in P group
Moura M.F., 2021 [70]	Prospective	RA + P (24) P (23) RA (19)	FM and RP	45 days	DAS28, RANKL, OPG, RANKL/OPG, and survivin in plasma and saliva	Reduction of salivary and serum RANK and survivin in RA + P group
Bıyıkoğlu B. et al., 2013 [122]	Single- centered intervention	RA + P (15) P (15)	FMS and RP	6 mo.	DAS28 IL-1β and TNF-α in serum and GCF	Reduction of DAS28 in RA +P group, reduction of GCF IL-1β and TNF-α in both group Positive correlation of DAS28 and GCF IL-1β with periodontal indexes
Salah S., 2023 [123]	Prospective interventional	RA + AS (25) RA + P (30) RA + AS + P (25)	FM and RP	6 mo.	DAS28, CIMT and FRS	Reduction of DAS28 and CIMT in RA + P group
Anusha D. et al., 2019 [124]	RCT	RA + P (15) RA + P (15) RA + P (15)	SRP SRP + MEC SRP + CHX	6 we.	CRP, ESR, APCAs, RF	Reduction in CRP, ESR, RF, and ACPAs was observed in all treatment groups, especially in SRP + MEC group
Elsadek M.F. et al., 2022 [125]	Clinical trial	RA + P RA + P	SRP SRP + Laser and PDT	12 we.	RF, IL-6, and TNF-α in GCF	Reduction of GCF, IL-6 and TNF-α in intervention group
Martu M.A. et al., 2021 [126]	CC	RA + P (26) P (26)	SRP SRP + PDT SRP + Laser	6 mo.	8-OhdG and 4-HNE in GCF	Reduction of GCF 8-OhdG and 4-HNE in both group

Aac: Aggregatibacter actinomycetem comitans, ACR20, ACR50, ACR70: American College of Rheumatology 20%, 50%, 70%, CPAs: Anti-citrullinated protein autoantibodies, AS: atherosclerosis, BASDAI: Bath Ankylosing Spondylitis Disease Activity Index, CarP: Carbamylated protein, CC: case–control, CHX: chlorhexidine, CIMT: carotid intima media thickness, CRP: C-reactive protein, DAS28: Disease Activity Score Calculator for Rheumatoid Arthritis, ESR: erythrocyte sedimentation rate, FM: full mouth, FMS: Full-mouth scaling, FRS: Framingham risk score, GCF: gingival crevicular fluid, GOHAI: Geriatric Oral Health Assessment Index, HAQ: health assessment questionnaire, 4-HNE: 4-Hydroxynonenal, IL-1: interleukine-1, IL-6: interleukine-6, IL-8: interleukine-8, IL-17: interleukine-17, LFM: leflunomide, MCP-1: monocyte chemoattractant protein-1, MEC: mouthwash containing essential oils and curcumin, MGI: modified gingival index, MMP-8: matrix metalloproteinase-8, MTX: methotrexate, MMP-3: matrix metalloproteinase-3, MMP-8: matrix metalloproteinase-8, MMP-9: matrix metalloproteinase-3, mo.: months, mono: monotherapy, NETs: neutrophil extracellular traps, N.R.: not reported, N.S.C.: no significant changes, NSAIDs: non-steroidal anti-inflammatory drugs, OHIP-14: Oral Health Impact Profile-14, 8-OhdG: 8-hydroxy-2′ -deoxyguanosine, OHI: oral hygiene index, OPG: osteoprogenin, P: periodontitis, PA: psoriatic arthritis, PAI-2: PAI-2: plasminogen activator inhibitor-2, PBI: papillary bleeding index, PD: probing depth, Pg: Porphyromonas gingivalis, PDT: photodynamic therapy, PGE2: prostaglandin E2, PIBI: periodontal inflammation burden index, PISA: periodontal inflamed surface, PCR: plaque control record, PI: plaque index, PL: plaque levels, PPD: periodontal probing depth, RA: rheumatoid arthritis, RANKL: receptor activator of nuclear factor-KB ligand, RCT: randomized controlled trial, RF: rheumatoid factor, RP: root planning, SDAI: Simple Disease Activity Index, SF-36: Short Form Health Survey 36, SpA: spondyloarthritis, SRP: scaling and root planning, SS: supragingival scaling, Td: Tannerella denticula, Tf: Tannerella forsythia, TIMP-1: tissue inhibitor of MMPs, TNF-α: tumor necrosis factor-α, t-PA: tissue plasminogen activator, VAS: Visual Analogue Scale, VPI: visual plaque index, we.: weeks.

**Table 3 ijms-24-17228-t003:** Reduction of cytokines and acute phase proteins after non-surgical periodontal treatment (NSPT).

Biological Sample	Cytokines and Inflammatory Mediators Impaired by NSPT
Serum	CarP, IL-6, NETs, RANKL, TNF-α, survivin, prolactin
GCF	IL-1β, IL-6, MMP-8, PAI-2, PGE2, t-PA, TNF-α, prolactin
Saliva	RANKL
Synovial fluid	Prolactin

CarP: Carbamylated protein, GCF: gingical creviculal fluid, IL-1β: interleukine-1 β, IL-6: interleukine-6, RANKL: receptor activator of nuclear factor-KB ligand, MMP-8: matrix metalloproteinase-8, PAI-2, NETs: neutrophil extracellular traps, plasminogen activator inhibitor-2, NSPT: non-surgical periodontal treatment, PGE2: prostaglandin E2, t-PA: tissue plasminogen activator, TNF-α: tumor necrosis factor-α.

**Table 4 ijms-24-17228-t004:** Studies concerning the effect of synthetic disease-modifying anti-rheumatic drugs (sDMARDs) on periodontal disease.

Authors	Study Design	Study Sample (N.)	sDMARDs	Follow-Up	Periodontal Indexes	Outcomes
Kaczynski T. et al., 2019 [71]	CS	RA + P (35) P (35) Healthy (36)	MTX, LFM, +/− GCs	6 mo.	API, BOP, CAL, N. missing teeth, PD, PD ≥ 4 mm Salivary IL-6, IL-8, IL-17A, and TNF-α	Reduction of BPO, PD, salivary IL-6, IL-17, TNF-α in RA + P group
Jung G.U. et al., 2018 [72]	CC	RA + P (32) P (32)	MTX, HCQ, SSZ	4 we.	PI, GI, PD, CAL, BOP	Reduction of CAL and PD in RA + P group
Nik-Azis N.M. et al., 2021 [73]	CC	RA + P (103) OA + P (103)	MTX+/− SSZ, HCQ, LMF +/− GCs	3 mo.	DMFT, GI, PD, CAL	Reduction of CAL and lower PD in MTX-mono vs. MTX-combo Reduction of GI in GCs group
Ancuta C. et al., 2019 [74]	Prospective longitudinal	RA + P (21)	Baricitinib	24 we.	N. teeth, CAL, GI, PD	Reduction of PD, nr. PD ≥ 4 mm, GI, sites with BOP

API: Approximal Plaque Index, BOP: bleeding on probing, CAL: clinical attackmen loss, CC: case–control, CS: cross-sectional, DMFT: decayed, missing, and filled teeth, GI gingival index, GCs glucocorticoids, HCQ: hydroxychloroquine, IL-6: interleukine-6, IL-8: interleukine-8, IL-17: interleukine-17, LFM: leflunomide, mo.: months, MTX: methotrexate, OA: osteoarthritis, PD: probing depth, PI: plaque index, RA rheumatoid arthritis, SSZ sulfasalazine, TNF-α: tumor necrosis factor-α, we.: weeks.

**Table 5 ijms-24-17228-t005:** Studies concerning the effect of biological disease-modifying anti-rheumatic drugs (bDMARDs) on periodontal disease.

Authors	Study Design	Study Sample (N.)	bDMARDs	Follow-Up	Periodontal Indexes	Outcomes
Kobayashi T. et al., 2014 [75]	Comparative	RA + P (20)	Adalimumab + DMARDs, NSAIDs, CGs	3 mo.	GI, PD, CAL, sites with BPO, sites with plaques	Reduction of GI, sites with BOP, PD
Mayer Y. et al., 2012 [76]	Comparative	RA + P (20) Healthy (10)	Infliximab + DMARDs + GCs (10) DMARDs + GCs (10)	26 ± 8 mo.	PI, GI, PD, BOP GCF-TNF-α	Reduction of BPO, PD, CAL, GCF-TNF-α in RA + infliximab group
Schiefelbein R. et al., 2018 [77]	CC	RA + P (13) P (13)	Etanercept, adalimumab, golimumab	12 mo.	BOP, CAL, GBI, PCR, PD, sites with PD ≥ 5 mm	Reduction of BOP, PD, sites with PD ≥ 5 mm, extension, and severity of periodontitis in RA + P group
Kadkhoda Z. et al., 2016 [78]	Prospective	RA + P (36)	Etanercept	6 we.	BOP, GI, PD, OHI, GCF-TNF-α	Reduction of BOP, GI, GCF-TNF-α
Üstün K. et al., 2013 [79]	Longitudinal	RA + P (16)	Adalimumabinfliximab	1 mo.	BOP, CAL, GI, PD, and PI Salivary and GCF IL-1β, IL-8, and MCP-1	Reduction of salivary IL-8 and MCP-1
de Smit M.J. et al., 2021 [80]	Observational longitudinal	RA + anti-TNF and MTX (12) RA + MTX (14)	MTX, anti-TNF	3–6 mo. for anti-TNF 2 mo. for MTX	PISA Microbiological analysis of subgingival plaque and IgG anti-Pg	N.S.C. in PISA and prevalence of Pg
Ziebolz D. et al., 2018 [81]	CS	RA + bDMARDs, sDMARDs (168)	NSAIDs + GCs MTX, LFN MTX + anti-TNF, MTX + RTX, anti-IL6	6 mo.	CAL, BOP, PBI, PD Microbiological analysis of GCF	Elevation of BOP and PBI for MTX + anti-TNF vs. LFN, elevation of BOP for MTX + anti-TNF vs. MTX + RTX Difference in the prevalence of Pg, Td, and Fn in different medication subgroup
Äyräväinen L. et al., 2018 [82]	Prospective	RA + sDMARDs mono or combo +/− bDIMARs (81) Healthy (54)	sDMARDs mono or combo +/− bDIMARDs	6 mo.	BOP sites, CAL, PD ≥ 4 mm, number of teeth, PIBI, VPI Microbiological analysis of subgingival plaque	N.S.C. in periodontal indexes and prevalence of Pg
Coat J. et al., 2015 [83]	CS and longitudinal	RA + P (11) RA + P (10)	Rituximab Rituximab (>2 more than two courses of two infusions)	6 mo. 48 mo.	CAL, DMFT, MGI, PD, PI, PBI	Reduction of CAL, MGI, PD, PI after 6 mo., improvement of periodontal status after 48 mo.
Kobayashi T. et al., 2015 [84]	Longitudinal CC	RA + P (20) RA + P (40)	Tocilizumab Anti-TNF	3–6 mo.	BOP, CAL, GI, and PD, PL	Reduction of BOP, GI, PD. Reduction of CAL in tocilizumab group
Ancuta C. et al., 2021 [85]	Prospective	RA + P (51)	Tocilizumab	3–6 mo.	PI, GI, site of BOP, PPD, CAL	Reduction of GI, sites of BOP, PPD
Punceviciene E. et al., 2021 [16]	CC	RA + P (63) P (30)	bDMARDs +/− sDMARDs	N.R.	CAL, PPD, BOP, and BL	Reduction of BOP and BL

bDMARDs: biological disease-modifying anti-rheumatic drugs, BL: bone loss, BOP: bleeding on probing, CAL: clinical attackmen loss, combo: combination therapy, CC: case–control, CP: chronic periodontitis, CS: cross-sectional, DMFT: decayed, missing, and filled teeth, DMARDs: disease-modifying anti-rheumatic drugs, GCF: gingival crevicular fluid, GBI: gingival bleeding, GI gingival index, GCs glucocorticoids, HAQ: Health Assessment Questionnaire, IL-6: interleukine-6, IL-8: interleukine-8, IL-17: interleukine-17, LFM: leflunomide, MMP-8: matrix metalloproteinase-8, MTX: methotrexate, MGI: modified gingival index, mo.: months, MCP-1: monocyte chemoattractant protein-1, mono: monotherapy, N.R.: not reported, N.S.C.: no significant changes, NSAIDs: non-steroidal anti-inflammatory drugs, OHI: oral hygiene index, P: periodontitis, PA: psoriatic arthritis, PBI: papillary bleeding index, PD: probing depth, Pg: Porphyromonas gingivalis, PIBI: periodontal inflammation burden index, PISA: periodontal inflamed surface, PCR: plaque control record, PI: plaque index, PL: plaque levels, PPD: periodontal probing depth, RA rheumatoid arthritis, Td: Tannerella denticula, TIMP-1: tissue inhibitor of MMPs, TNF-α: tumor necrosis factor-α, VPI: visual plaque index, we.: weeks.

## Data Availability

Not applicable.

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
