# Peer review of "The Effects of Periodontal Treatment on Rheumatoid Arthritis and of Anti-Rheumatic Drugs on Periodontitis: A Systematic Review"

_ijms, 2023, doi:10.3390/ijms242417228_

Round 1
Reviewer 1 Report
Comments and Suggestions for Authors
This article is an extensive review of RA and periodontitis. Some comments
It is better to add “Systematic review” to the title.
In the Abstract, please add more results.
In the Introduction, adding more molecular dynamics of periodontal treatment and dental treatments is better.
https://doi.org/10.3390/molecules25204650
https://doi.org/10.3390/jfb13010015
Tables 3 and 4 need to be edited. Add the text from the Caption to the footnotes.
Table 4 is too long, is it possible to find a way to break the Table into 2 headings?
Add limitations of this research.
The conclusion is long, please reduce the conclusion and include the main points only.
Comments on the Quality of English LanguageMinor English correction is needed.
Reviewer 2 Report
Comments and Suggestions for Authors
In the current review the authors evaluated the effects of non-surgical periodontal treatment on rheumatoid arthritis and the impact of disease modifying anti-rheumatic drugs on periodontitis. The authors state that the restoration of periodontium could be another key strategy to reduce rheumatoid arthritis severity, just as anti-rheumatic therapy can also positively affect periodontium status.
Some minor comments:
1. Point 2.6. Quality Assessment, line 127-28: It would be interesting to add which are the seven evaluated points.
2. Table 2: The follow-up during non-surgical periodontal treatment varies from 45 days to 6 month. In my opinion it’s a great difference between 45 days and 6 month. The same observation is valid for the data recorded in Table 4.
3. Pg 11, line 328: please add which essential oils does the mouthwash contain?
4. Pg 11: Curcumin is obtained from Curcuma plant, so is not a chemical compound it/s a natural compound (see please line 320)
5. Pg 11-12, Give please more details concerning lines 333-340.
6. An explanation of abbreviations is necessary when they first appear in the text.
Comments on the Quality of English LanguageMinor editing of English language is required.
